# An Isolated *Arthrobacter* sp. Enhances Rice (*Oryza sativa* L.) Plant Growth

**DOI:** 10.3390/microorganisms10061187

**Published:** 2022-06-09

**Authors:** Geeta Chhetri, Inhyup Kim, Minchung Kang, Yoonseop So, Jiyoun Kim, Taegun Seo

**Affiliations:** Department of Life Science, Dongguk University-Seoul, Goyang 10326, Korea; geetachhetri123@yahoo.com (G.C.); duckling91@dgu.ac.kr (I.K.); melissak2023@gmail.com (M.K.); sts5552@naver.com (Y.S.); ginapd@daum.net (J.K.)

**Keywords:** culture-dependent, IAA, siderophore, _L_-tryptophan, PGP, bioremediation, phytopathogen

## Abstract

Rice is a symbol of life and a representation of prosperity in South Korea. However, studies on the diversity of the bacterial communities in the rhizosphere of rice plants are limited. In this study, four bundles of root samples were collected from the same rice field located in Goyang, South Korea. These were systematically analyzed to discover the diversity of culturable bacterial communities through culture-dependent methods. A total of 504 culturable bacteria were isolated and evaluated for their plant growth-promoting abilities in vitro. Among them, *Arthrobacter* sp. GN70 was selected for inoculation into the rice plants under laboratory and greenhouse conditions. The results showed a significantly positive effect on shoot length, root length, fresh plant weight, and dry plant weight. Moreover, scanning electron microscopic (SEM) images demonstrated the accumulation of bacterial biofilm networks at the junction of the primary roots, confirming the root-colonizing ability of the bacterium. The strain also exhibited a broad spectrum of in vitro antimicrobial activities against bacteria and fungi. Here, we first report the rice plant growth-promoting ability of the *Arthrobacter* species with the biofilm-producing and antimicrobial activities against plant and human pathogens. Genome analyses revealed features attributable to enhance rice plant growth, including the genes involved in the synthesis of plant hormones, biofilm production, and secondary metabolites. This study revealed that the rhizobacteria isolated from the roots of rice plants have dual potential to be utilized as a plant growth promoter and antimicrobial agent.

## 1. Introduction

Rice (*Oryza sativa*) is the staple crop for more than half of the world’s population, including South Korea [1]. With the increasing world population, the demand for rice is expected to increase and therefore there is an immense pressure for higher production to meet the food demands. Today, our challenge is to increase crop productivity at a faster rate than population growth; therefore, interest has increased recently in the beneficial rhizobacteria associated with cereals. Several studies have clearly demonstrated the positive and beneficial effects of plant growth-promoting bacteria (PGPB) on the development and yield of different crops, in particular, wheat and rice in different environments under variable ecological conditions [2,3,4,5]. A major problem associated with rice production is the substantial use of chemical fertilizers, which can cause negative impacts on both human health and the environment [6]. The *Arthrobacter* species are known to utilize organic and inorganic compounds as a substrate of metabolism; thus, acting as a tool for bioremediation. Therefore, most of their studies are performed with regards to their bioremediation as compared to PGPB properties. Different strains are involved in the degradation of heterocyclic organosulfur compounds, halo aromatics, herbicides, and pesticides from the environment [7,8,9]. The PGPB exert beneficial effects on plants through direct and indirect mechanisms. They promote plant growth directly usually by either facilitating resource acquisition or modulating plant hormone levels, and indirectly by decreasing the inhibitory effects of various pathogenic agents on plant growth and development [10]. The indirect mechanism acts as an antibiotic against the plant pathogens, thereby, providing biological control activity, and the direct mechanism includes siderophore production, plant hormone production, phosphorus solubilization, etc. Additionally, PGPB strains can improve the stress tolerance of plants, which acts to moderate the climate change [11].

Natural endophytic bacteria associated with the plants have become a promising alternative to chemical fertilizer, due to their plant growth-promoting abilities. The PGPB isolated from rice fields were shown to increase the plant height, root length, and dry matter production of the shoots and roots in rice seedlings [7]. A plethora of PGPB from the genera *Pseudomonas*, *Bacillus*, *Enterobacter*, *Acinetobacter*, *Arthrobacter*, *Burkholderia*, *Paenibacillus*, *Agrobacterium*, *Lysinibacillus,* and *Flavobacterium* were reported to give positive effects on the rice plant’s health and growth [12].

This is a continuation study from our previous work, where we isolated many species from different genera associated with the roots of the rice plant [13,14,15]. We tested for IAA production and found that most of the bacteria isolated from the roots of the plant produce indole [15]. Previous studies also showed that most of the rhizosphere bacteria are able to produce IAA [3].

Further, to understand the function of bacterial communities in the roots as PGPB, we collected the root samples of *Oryza sativa* (L.) and performed a spread plate technique to isolate the cultured bacteria. We performed PGPB tests for each colony based on its high IAA levels for inoculation (in vitro).

Previous studies showed that the PGPB species should be good root colonizers and ought to produce secondary metabolites, which make them good biocontrol agents against plant pathogens [16,17]. In our study, we also found that the selected isolate was able to produce biofilm under static conditions. The SEM observation also confirmed this result by observing a network of biofilm at the junction of the primary root.

In this study, we isolated 504 strains through culture-dependent methods and tested for the PGPB traits for each strain. Based on its high IAA value, PGPB traits, and antagonism towards pathogenic microorganisms, *Arthrobacter* sp. GN70 was selected for inoculation in rice plants, under lab and greenhouse conditions. Strain GN70 successfully attached to the roots, produced fibril matrix, and enhanced the rice plant growth, lateral roots, and weight. The strain also showed antagonistic effects towards plant and human pathogens. As far as we know, our study was the first to demonstrate that the *Arthrobacter* species promoted growth of the host rice plant both under lab and greenhouse conditions. The strain GN70 could also contribute to their healthy growth by biocontrol activity.

## 2. Materials and Methods

### 2.1. Isolation and Identification

For culture-dependent sampling, fresh roots from one bundle of *Oryza sativa* (L.) were collected from rice fields located in Goyang, Gyeonggi-do, South Korea. The samples were collected from four locations based on a specific distance. The samples were placed in sterile vials and taken immediately to the laboratory for screening. Root samples were disinfected two times with 70% ethanol for 30 s and washed three times with distilled water. At last, sterile filter paper was used to absorb the water. Subsequently, the samples (10 g) were ground using sterile tweezers and a mortar, as described previously [18]. An aliquot of 100 μL of the sample suspensions were spread onto Reasoner’s 2A (R2A, BD Difco, Franklin Lakes, NJ, USA) agar plates and subsequently incubated at 28 °C for 5 days. Colonies were subsequently selected to purify by streaking. All of the strains were stored in 25% glycerol (w/v) at −80 °C. The 16S rRNA gene sequence of all of the strains were amplified using the universal bacterial primer sets 27F, 518F, 805R, and 1492R [18], and its sequencing was performed by Solgent Co. Ltd. (Daejeon, Korea). The almost complete sequences of the 16S rRNA genes (>1400 bp) were assembled using the SeqMan software (DNASTAR Inc., Madison, WI, USA). The 16S rRNA gene sequencing results were compared in EzBioCloud.net (http://www.ezbiocloud.net, accessed on 23 May 2022) [19] and BLAST searches from the NCBI database [20].

### 2.2. Plant Growth-Promoting Traits

The 504 isolates collected from the culture-dependent screening were tested for their plant growth-promoting traits, including indole-3-acetic acid (IAA) production, siderophore production, nitrogen fixation, and phosphate solubilization. Among the isolates, the 30 bacterial isolates with the highest IAA production were selected for further comparison.

#### 2.2.1. Detection and Quantification of Indole-3-Acetic Acid Production

The bacterial culture supernatant was evaluated using a colorimetric method to classify auxin-producing isolates. The isolates were incubated in R2A broth in the presence of 0.1% _L_-tryptophan as a precursor at 30 °C and 180 rpm for 72 h in the dark. Following incubation, the bacterial cells were removed from the culture medium by centrifugation (16,780× *g* for 5 min). Bacterial supernatant aliquots of 100 μL were mixed with Salkowski’s reagent (98 mL 35% perchloric acid and 2 mL 0.5 M FeCl_3_) in a 1:2 ratio [21]. The mixture was left to stand at room temperature for 30 min before the absorbance was measured at 530 nm. The concentration of IAA in each culture medium was determined by comparison against an IAA standard curve.

#### 2.2.2. Siderophore Production

The siderophore production was assessed through the blue agar chrome azurol S (CAS) assay, where the bacterial isolates were spot inoculated onto the agar and incubated at 30 °C for a week [22]. When positive for siderophore production, the agar surrounding the bacterial colonies would change color from blue to yellow/orange.

#### 2.2.3. Screening for Putative Nitrogen Fixation

The bacterial isolates were tested for nitrogen fixation with Jensen’s nitrogen-free agar media and incubated at 30 °C for a week after inoculating the strains onto individual plates. Luxuriant growth of the inoculated colonies indicated positive results for nitrogen fixation [23].

#### 2.2.4. Phosphate Solubility

To test for phosphate solubility, the isolates were spot-inoculated onto Pikovskaya’s agar, where calcium phosphate was utilized as a source for phosphorus [24]. Clear zone formations in the vicinity of the bacterial colonies indicated phosphate solubilization.

### 2.3. Biofilm Formation

Based on the results, *Arthrobacter* sp. GN70 was selected to test the ability for bio film production. The GN70 was inoculated from fresh R2A agar plates into LB broth and incubated for 48 h at 30 °C under static conditions, as described in our recent work [25].

### 2.4. Antimicrobial and Antifungal Test

Antibacterial and antifungal tests were investigated against the following 15 strains: *Pseudomonas aeruginosa* ATCC 6538; *Staphylococcus aureus* subsp*. aureus* ATCC 6538; *Pantoea agglomerans* KACC 10054; *Xanthomonas campestris* pv. *campestris* KACC 10377; *Staphylococcus epidermidis* KACC 13234; *Bacillus subtilis* subsp. *subtilis* KACC 16747; *Aspergillus niger* KACC 42850; *Botrytis cinerea* KACC 40573; *Colletotrichum gloesporioides* KACC 40003; *Pythium spinosum* KACC 45737; *Fusarium proliferatum* KACC 44025; *Fusarium graminearum* KACC 46577; *Fusarium solani* KACC 44891; *Fusarium semitectum* KACC 41036; and *Alternaria alternata* KACC 44413.

Here, only *Arthrobacter* sp. GN70 was selected to evaluate the antimicrobial activity of pathogenic microbes. The antimicrobial activity was detected using the spotting method for pathogenic microbes. For this purpose, 24-h-old pathogenic bacteria were spread onto an R2A plate and the freshly cultured GN70 strain was spotted in the middle of the plate. The plate was incubated at 30 °C for 48 h.

For the antifungal activity, the fungi were streaked onto Corn Meal Agar (CMA, MB cell) and Potato Dextrose Agar (PDA, MB cell) and the freshly cultured GN70 strain was spotted into the four corners of the plates. The plates were then incubated for seven days at 30 °C and 25 °C, respectively.

### 2.5. Rice Seed Sterilization, Germination, and Innoculation (In Vitro)

Healthy seeds of rice (IT267968) were received from the Korean Agricultural Culture Collection (KACC). Similar sized rice seedlings were de-husked by hand, surface sterilized with 70% ethanol, and then soaked in 5% Clorox and Tween 20 detergent for 5 min. Finally, the seeds were washed five times with sterilized water. Rice seeds were placed in a sterilized moist filter paper and allowed to germinate in the dark, in an incubator set at 28 °C. After 48 h of growth, the seeds were left to dry in a sterile petri dish inside a clean bench for two hours. Thirty sterilized rice seeds were covered with a fresh bacterial suspension of *Arthrobacter* sp. GN70 (OD_600_ = 0.5) for 5 h. During this, gentle mixing occurred to allow bacterial colonization on the rice seeds. Thirty seeds were also treated with sterilized distilled water as a negative control. Ten seeds were placed on each plate with Murashige and Skoog (w/v) supplemented with 0.7% agar medium (Murashige and Skoog, 1962). Triplicates of each treatment were maintained and incubated in a plant growth chamber on a 14 h light and 10 h dark cycle for 14 days at 25 °C. Photographs were taken after 7 and 14 days. After 14 days, the rice plants’ growth parameters were measured.

### 2.6. Pot Experiment Assay

For the pot experiment, surface-sterilized healthy rice seeds were selected. Sixty seeds for the non-inoculated (control) and sixty seeds inoculated with GN70 were used and planted into pots (20 cm × 15 cm × 10 cm), filled with sterilized field soil. These were then kept in the greenhouse of Dongguk University, Goyang, South Korea. The experimental soil was brought from the nearest rice field. The control pot was irrigated with water only, while the inoculated pot was irrigated with water (daily) and with 20 mL of GN70 suspension (OD_600_ = 0.5) every week. After four weeks, the plants were harvested, and the roots were washed thoroughly with running tap water to remove the adherent soil in the plants. The length of the shoots and roots were recorded. In addition, the fresh weight and dry weight of each plant was also measured. Data was statistically analyzed using the *t*-test.

### 2.7. Plant Root Colonization

Two independent tests were completed to investigate whether *Arthrobacter* sp. GN70 was able to colonize the roots of rice plants. Firstly, seven days after inoculation, surface sterilization was performed by submerging the roots in 70% ethanol for 5 min and absolute ethanol for 1min. The samples were air-dried and placed in a 6% (*v*/*v*) commercial sodium hypochlorite solution for 2 min. Subsequently, excessive sodium hypochlorite solution was removed by rinsing the samples five times in sterilized distilled water. The sterilized plant tissues were then macerated with 0.85% NaCl with sterile mortars to obtain an aqueous extract. A volume of 100 μL of the extracts was serially diluted and plated onto the R2A agar for 3 days at 30 °C, as described previously [26]. To ensure the purity of the strain, a sample was sub-cultured on R2A agar plates and sent to Solgent Co. Ltd. (Daejeon, Korea) for 16S rRNA gene sequence analysis.

For the second test, an SEM observation was performed following 14 days of incubation. Here, the roots were cut and fixed overnight in 1 mL of 2.5% glutaraldehyde, diluted using phosphate buffer at pH 7.4, in accordance with our previous studies [25]. The sample was dehydrated through consecutive washes with 30, 50, 60, 70, 80, 90, and 100% (*v*/*v*) absolute ethanol. After drying with hexamethyldisilazane, the root samples were coated with platinum using ion sputtering equipment (15 nm; EM ACE200, Leica, Wetzlar, Germany) and observed under a scanning electron microscope (FESEM, Sigma, St. Louis, MO, USA).

### 2.8. Genome Annotation and Analysis of Genes Involved in PGPB

The genome sequence of strain GN70 was determined using the Illumina Hiseq 4000 platform with 150 bp paired-end reads, according to the manufacturer’s protocols. Genome assemblies were generated using the SPAdes program, as described previously [27]. Genome annotation was performed by means of the NCBI Prokaryotic Genome annotation pipeline (PGAP) [20]. The CheckM bioinformatics tool was used to assess the genome contamination and completeness (https://ecogenomics.github.io/CheckM, accessed on 23 May 2022) of strain GN70 [28]. Functional annotation was conducted by eggNOG (evolutionary genealogy of genes: Non-supervised Orthologous Groups) 4.5 database [29]. To investigate this antagonistic activity, the biosynthetic potential of GN70 was evaluated by using antiSMASH 6.0.0 to predict both the characterized and unknown functions of the secondary metabolites [30].

## 3. Results

### 3.1. Isolation and Identification

All of the colonies were collected, cultured, and subsequently purified several times by repeated streaking to obtain pure colonies. A total of 504 strains were isolated. All of the selected strains were stored in 25% glycerol (*w*/*v*) at −80 °C. Prior to evaluating the functionality of the isolates, incubation was confirmed at various temperatures ranging from 20 to 37 °C to check whether the isolates were growing in the R2A.

### 3.2. Plant Growth Promoting Traits

A total of 504 isolates were isolated by the culture-dependent method and, among them, 60% of the strains were able to produce IAA. A total of 30 strains with high IAA were selected for further studies. From the 30 isolates, 60% of the isolates were positive for siderophore production. These isolates belonged to genera such as *Sphingomonas*, *Ideonella*, *Burkholderia*, and *Agromyces*. The *Arthrobacter* species, designated as *GN70*, produced the highest amount of IAA (50.3 µg/mL) (Table 1). The phosphate solubility test results were positive in only three isolates, all of which belonged to the genus *Burkholderia,* while the nitrogen fixation capabilities were present in 40% of the isolates. Isolate P2, which was identified as the species *Burkholderia vietnamiensis* LMG 10,929 (100% similarity), was the only isolate to show positive results in all four of the traits. Meanwhile, seven of the isolates displayed positive results for three traits, in which two belonged to the genus *Burkholderia* and two belonged to the genus *Ideonella*. The plant growth-promoting traits of the isolates are summarized in Table 1.

### 3.3. Biofilm Formation

The cells formed biofilms, which were confirmed by the presence of floating pellicles (visible film) in the liquid media, while the broth remained transparent (Figure 1). The ability to form biofilms both enhances bacterial survival and also enhances plant growth through the various PGPR-associated mechanisms [31,32]. Within the biofilm, extracellular polymeric substances (EPS) are comprised of polysaccharides, extracellular DNA, proteins, and lipids, and are distributed between the cells in a non-homogeneous pattern [31]. They provide a microenvironment that holds water and dries more slowly than its surrounding environment; thus, protecting the bacteria and roots of plants against desiccation [11]. The biofilm produced by PGPR produces remarkably higher amounts of antimicrobial compounds, leading to the suppression of phytopathogens [32]. The GN70 can be utilized as plant growth promoters, suppressors of plant pathogens, and alleviators of water-deficit stress.

### 3.4. Antimicrobial and Antifungal Test

Among the 15 pathogens, *Arthrobacter* sp. GN70 showed antagonism with seven of the pathogenic strains (Table 2). The antifungal activity of GN70 showed an inhibitory region only in the F. *proliferatum* KACC 44025, which mainly affects the rice plant with rice spikelet rot disease. These findings suggest that GN70 could aid in the prevention of rice spikelet rot disease.

### 3.5. Innoculation of Arthrobacter sp. GN70

After 7 and 14 days of incubation, the rice plants illustrated discernible differences compared to their respective control groups (Figure 2). The GN70 showed a significant effect on the growth of the rice plant, which was noticeable in the shoot and root system. Previous studies showed that plant growth-promoting rhizobacteria (PGPR) reduced the growth of the main root but increased the number and length of the lateral roots alongside stimulating the root hair elongation. This enhanced the uptake of water and nutrients culminating in increased plant growth and development [33,34]. In our study, we also found there was not much difference in the length of the roots after 7 days of incubation between the control and inoculated groups. However, the lateral root development increased in the plants inoculated with the GN70 as compared to the control in Figure 2c,f. When compared to the control groups, the seeds inoculated with the GN70 increased in shoot length, root length, fresh plant weight, and dry plant weight by 57.7%, 26.7%, 89.6%, and 97.1%, respectively. The different growth parameters of the non-inoculated and inoculated rice plants are illustrated in Figure 3.

### 3.6. Innoculation Effects in Pot Experiment

We performed a pot experiment, during August for only four weeks. Unfortunately, the rice plant growth experiment could not be performed over an enhanced duration, because rice is very sensitive to temperature, and the winter in South Korea begins during late August. However, the results from the four weeks illustrated that the rice plants treated with GN70 were taller as compared to the untreated control plants. Moreover, the GN70 treatment significantly enhanced the shoot and root length, alongside the dry weight of the shoots and roots. In addition, the leaves of the treated plants were also wider than the controls (Figure 4). Based on the *t*-test, the seeds inoculated with GN70 increased in shoot length, root length, fresh plant weight, and dry plant weight by 143.5%, 83.5%, 112.1%, and 256.7%, respectively. The different growth parameters of the non-inoculated and inoculated rice plants are illustrated in Figure 5.

### 3.7. Plant Root Colonization

The PGPR are a group of bacteria capable of actively colonizing the root system of plants and improving their growth and yield. Previous studies showed that *Arthrobacter*, *Alcaligenes*, *Azospirillum*, *Azotobacter*, *Bacillus*, *Burkholderia*, *Enterobacter*, *Klebsiella*, *Pseudomonas,* and *Serratia* are the most widely known microbial genera that have shown the ability to colonize plant rhizospheres and enhance plant growth [35]. The colonization of bacteria was tested by observing the successful re-isolation of bacteria. Our results obtained by the 16S rRNA gene sequencing showed that the strains, obtained inside the root surface, belong to the genus *Arthrobacter* and are 100% similar to the GN70. Using the SEM observations, we investigated the colonization of the rice root surface by *Arthrobacter* sp. GN70. As shown in Figure 6a, the cells of GN70 were observed at the junction of the primary roots, suggesting that GN70 could attach to the rice root and colonize it. Figure 6a shows that some of the bacteria produce exopolysaccharides, while the cells are embedded in a net formed by a polymeric matrix. Conversely, no such structure was observed in the control plants (Figure 6b). Previous studies have shown that bacterial EPS create a microenvironment that holds water and dries up more slowly, thereby protecting the bacteria and plant roots against extreme desiccation [11,25]. In our study, *Arthrobacter* sp. GN70 also produces EPS, which may hold water or increase the water-holding capacity in the soil to assist the bacteria and the plant roots under water-deficit stress.

### 3.8. Genome Annotation and Analysis of Genes Involved in PGPB

The comparison of the 16S rRNA gene sequence of strain GN70 in NCBI GenBank and EzBio-Cloud database revealed that strain GN70 shared 100% similarity with *Arthrobacter terricola* JH1-1^T^ (Table 2). The assembled draft genome of strain GN70 (JAHHIO000000000) contained 81 contigs, with a total length of 5.53 Mbp and an N50 length of 154,926 bp. Functional genes related to plant growth-promoting activity were also identified. The bioinformatics tool, CheckM, provided an assembled genome completeness of 99.26% and contamination of less than 1.0%. The clusters of the orthologous genes’ data obtained from the eggNOG analysis in relation to the GN70 strain genome, denoted that a total of 4760 genes were assigned to 24 functional categories. Among the obtained functional groups (the function unknown cluster is skipped), the cluster for E (amino acid transport and metabolism), G (carbohydrate transport and metabolism), and K (transcription) were the most highly represented categories. Other functional categories are presented in Figure 7.

Functional genes related to plant growth-promoting activity were identified. Various genes that are putatively involved in siderophore production, auxin biosyn thesis, nitrogen fixation, and phosphate solubilization were found (Appendix A).

*Arthrobacter* species are mostly involved in nitrogen fixation; however, we only found one set of gene *nifU* that confers nitrogen fixation. Other sets of genes for nitrogen fixation were not found. This could be due to the gaps in the draft genome sequence of the strain GN70. They also contain four gene clusters encoding acetolactate synthase (*ilvCDN*), which is involved in the synthesis of the plant growth promotion through inducing systemic resistance, and biocontrol of phytopathogens [36]. Genome annotation also predicted eight gene clusters for putative cold-shock proteins which are supposed to play an important role in low temperature conditions. Moreover, the heat-shock genes *dnaJ*, *dnaK*, *groEL*, *groES*, *htpX*, *hrcA,* and *grpE* were also found. These genes play a role in the protection of cellular oxidative stress caused by salt stress [37]. Plant growth-promoting bacteria reduce the salt stress through the synthesis of 1-aminocyclo-propane-1-carboxylate (ACC) deaminase or its homolog, D-cysteine desulfhydrase, which is encoded by *acdS* or *dcyD*, respectively [38]. These genes were not found in strain GN70. Two gene clusters for nitrogen regulatory protein P-II were also found, that control the level of nitrogen by regulating glutamine [39]. The GN70 also contained multiple genes involved in the antioxidant response, such as peroxidases, catalases, superoxide dismutase, and glutathione peroxidase. The *Arthrobacter* sp. GN70 further encodes genes for riboflavin synthase, which catalyzes the final reaction of riboflavin synthesis. Riboflavin stimulates plant growth and functions as the elicitor/protectant in the plant’s defense [40]. The NCBI annotation also revealed the two gene clusters for phenazine biosynthesis protein, which are associated with the production of antifungal compounds [36]. Previous studies showed that phenazine-producing bacteria are prevalent in the rhizosphere of rice and wheat [41,42,43]. Details of the genes are provided in Appendix A.

AntiSMASH revealed that GN70 harbored biosynthetic gene clusters (BGCs) coding for beta-lactone (microansamycin; 7% similarity), phosphonate (dehydrophos; 11%), non-alpha poly-amino acids such as e-Polylysin (NAPAA) (stenothricin; 31%), butyrolactone, bottromycin, post-translationally modified peptides (RiPPs), type 3 polyketide synthase (T3PKS), thiopeptide, linear azol (in)e-containing peptides (LAP) and amglycyl (cetoniacytone; 9%). Notably, four of the gene clusters shared similarities with anticancer and antimicrobial compounds, such as *cetoniacytone A*, *stenothricin*, *microansamycin,* and bottromycin. However, the levels of similarity were low, which suggest the novelty of the possible metabolites from those predicted gene clusters. The synthesis of these secondary metabolites might help to inhibit the growth of the pathogens.

## 4. Discussion

Plants depend on soil microorganisms as much as they rely on sunlight and water to grow. These rhizospheric bacteria, or PGPB, assist plants by acquiring nutrients through mechanisms such as nitrogen fixation, siderophore production, and phosphate solubilization, or by producing plant hormones that stimulate plant growth [44]. Additionally, rhizospheric bacteria may even provide protection against harmful pathogens [45]. A derivative of indole, IAA, is the most well-known phytohormone of the auxin class and is responsible for its role in cell division, senescence, and initiation in roots, flowers, and leaves [46,47]. IAA is also believed to aid rice plants in developing deep root systems, which help the plants in surviving arid conditions [48]. The PGPB also support plant growth by making iron (Fe) available to the plants, because it is normally present in its poorly soluble, oxidized form [49]. Since Fe in plants is used in important processes such as chlorophyll biosynthesis and photosynthesis, the PGPB are needed as they make soluble Fe available. This is then utilized in the production of iron-solubilizing organic acids or iron-chelating siderophores [50,51]. Additionally, some of the PGPB can limit the growth of phytopathogens by secreting siderophores and by sequestering available Fe from the environment [52]. Phosphorus (P) is another essential micronutrient that is difficult to utilize as it is typically in its insoluble form. However, phosphorus-solubilizing microorganisms can excrete H^+^ ions or organic acids that hydrolyze the recalcitrant forms of P, which can then be assimilated by plants [2]. Furthermore, nitrogen (N), a crucial element that is required in nucleic acid and protein synthesis, is fixed from the surrounding atmosphere with the assistance of the PGPB. By producing nitrogenase, the bacteria reduce the nitrogen gas (N_2_) to ammonia (NH_3_), an organic form that is available for plants [53].

The *Arthrobacter* sp. GN70 exhibited negative results for phosphate solubility. However, the strain exhibited numerous other growth-promoting traits, including high level IAA production (50.3 µg/mL), siderophore production, and nitrogen fixation. As such, the GN70 strain was selected for further analyses to assess its capabilities in promoting growth in rice plants. For our experiments, the rice plant was chosen as a model, due to the plant being the isolation source of the *Arthrobacter* sp. GN70. Since the isolation source was the roots of the rice plant, we were interested to discover whether this strain could promote the growth of rice plants and act as a biocontrol agent to combat plant pathogens. Our study showed that *Arthrobacter* sp. GN70 carries the potential to increase rice plant productivity, including lateral root growth and the weight of the plant under both lab and greenhouse conditions.

Rhizobacteria possess the ability to inhibit the growth of pathogenic bacteria and fungi [54]. Similar results were found in our study, where the *Arthrobacter* sp. GN70 exhibited extremely good antimicrobial activity against plant pathogens, such as *X*. *campestris* pv*. campestris* KACC 10377 and *P*. *agglomerans* KACC 10054. The *X. campestris* pv*. campestris* is an economically important bacterial plant pathogen worldwide because it causes black rot disease that devastates many cultivated cruciferous crops, through the production of V-shaped necrotic lesions on the foliar margins and blackened veins [55]. The *P*. *agglomerans* KACC 10054 was reported for leaf blight on rice in South Korea [56].

The strain *G*N70 also inhibited the growth of the human pathogens *S*. *aureus* ATCC 15442, *S*. *epi dermidis* KACC 13234, and *B. subtilis*. Previous reports also showed the antagonistic activity of the *Arthrobacter* species against these pathogenic bacteria, except for *P*. *agglomerans* KACC 10054, *S*. *epidermidis* KACC 13234, and *B. subtilis* [36].

Moreover, strain GN70 demonstrated antifungal activity against *Fusarium prolifer atum* KACC 44025, which causes rice spikelet rot disease. The disease causes rice grains to rot, discolor, and results in deformations to the grains, alongside reductions in the grain yield [57]. This is the first report of the antifungal activity of *Arthrobacter* species against *F*. *proliferatum* KACC 44025. Genome annotation of strain GN70 also revealed gene clusters responsible for the synthesis of volatile organic compounds which potentially inhibit *F*. *proliferatum* KACC 44025. Previous reports showed that *A. agilis* UMCV2 produces volatile organic compounds that inhibit the growth of *Botrytis cinerea* in vitro, which is a necrotrophic fungus that affects many plant species [58]. Therefore, we also performed an antifungal activity analysis of the strain GN70 against *B*. *cinerea* KACC 40573. However, we did not discover any discernible difference to the control data.

The species of *Arthrobacter* genus have been isolated from diverse stressful environments, such as saline, drought, pollution, and low nutrient agricultural soils. These are a selection of the locations in which the *Arthrobacter* species are found to be already involved in protecting the plants from abiotic stresses and improving the plant nutrition, health, and yields [4,59,60]. Owing to these beneficial properties towards plants, they are an important member among the plant growth-promoting rhizobacteria [61]. Despite their importance towards plant health under stressful conditions, we did not find any evidence that they promote rice plant growth (in vitro), even though they are isolated from the plant rhizospheres. Most of the studies were performed based only on the genomic insight. In our study, the in vitro experiments showed that *Arthrobacter* sp. GN70 has the ability to enhance plant growth and also showed antagonist activity against pathogenic microbes. These results motivated us to identify the genomic information required to understand the plant growth mechanism and the synthesis of secondary metabolites. We identified the clusters of genes responsible for PGPB and biocontrol activity of the selected strain. Moreover, other members of the genus *Arthrobacter* have also been reported for PGPB and biocontrol activity [62].

The colonization of bacteria is very important for interactions between terrestrial plants and the PGPB. Therefore, we investigated the colonization of GN70 in roots. Through our experiments, we found that the selected strain was able to colonize the roots of rice plants. Interestingly, we detected network structures of EPS, which are responsible for the colonization of bacteria on the plant surface and tight cell attachment. The ability of GN70 to successfully colonize and persist in the roots of rice plants is an important trait for its possible use as a bio-inoculant in agriculture. The use of microbes that are symbiotic with plants to promote plant growth and control disease is potentially cost-effective and may result in reductions in the use of synthetic fertilizers and pesticides to cultivate crop plants [54].

## Figures and Tables

**Figure 1 microorganisms-10-01187-f001:**
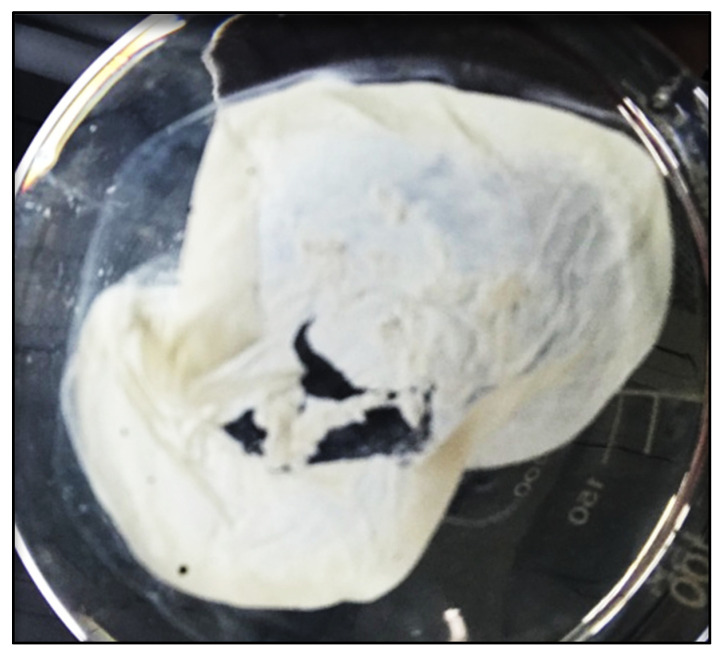
Floating biofilm formed in conical flask. *Arthrobacter* sp. GN70 was grown in LB broth under static condition for 3 d at 30 °C.

**Figure 2 microorganisms-10-01187-f002:**
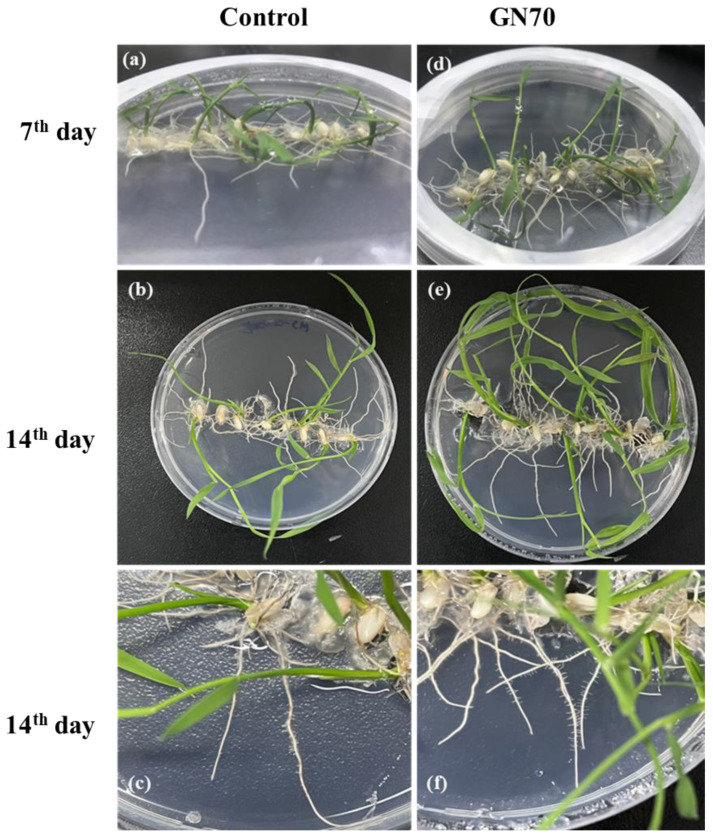
Effects of *Arthrobacter* sp. GN70 inoculation on the growth of rice seedlings after 7 and 14 days of incubation. (**a**–**c**) representative images of seedlings growing on control plates; (**d**–**f**) are seedlings inoculated with GN70. The experiment was repeated three times. Figure (**c**,**f**) show the comparison of lateral root development.

**Figure 3 microorganisms-10-01187-f003:**
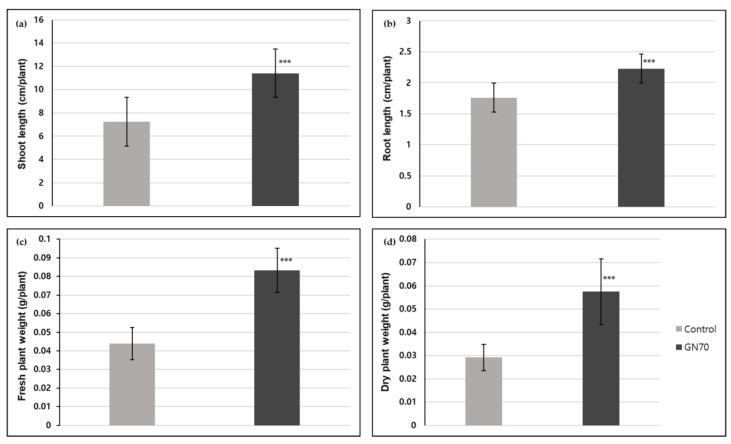
Effect of GN70 inoculation on rice seeds (IT267968) in vitro after 14 days of incubation. On average, shoot length (**a**) and root length (**b**) increased by 57.7% and 26.7%, respectively. Fresh plant biomass (**c**) increased by 89.6% and dry plant biomass (**d**) by 97.1% (average from 60 explants ± SD, *** *p* ≤ 0.01).

**Figure 4 microorganisms-10-01187-f004:**
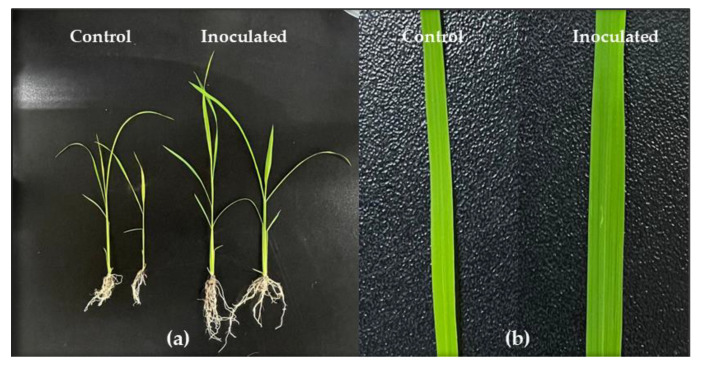
The effect of GN70 on the growth promotion of rice plants under greenhouse conditions. (**a**) Seeds were used as control; (**b**) seeds were pretreated with bacterial inoculation and then transferred into the soil pot. Measurement was taken after 4 weeks of germination.

**Figure 5 microorganisms-10-01187-f005:**
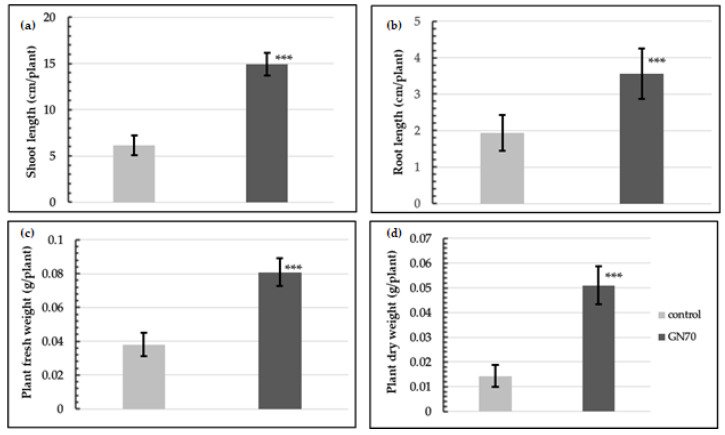
Effect of GN70 inoculation on rice seeds (IT267968) in pot experiment in green house after 4 weeks of growth. On average, shoot length (**a**) and root length (**b**) increased by 143.5% and 83.5%, respectively. Fresh plant biomass (**c**) increased by 112.1% and dry plant biomass (**d**) by 253.2% (average from 60 explants ± SD, *** *p* ≤ 0.01).

**Figure 6 microorganisms-10-01187-f006:**
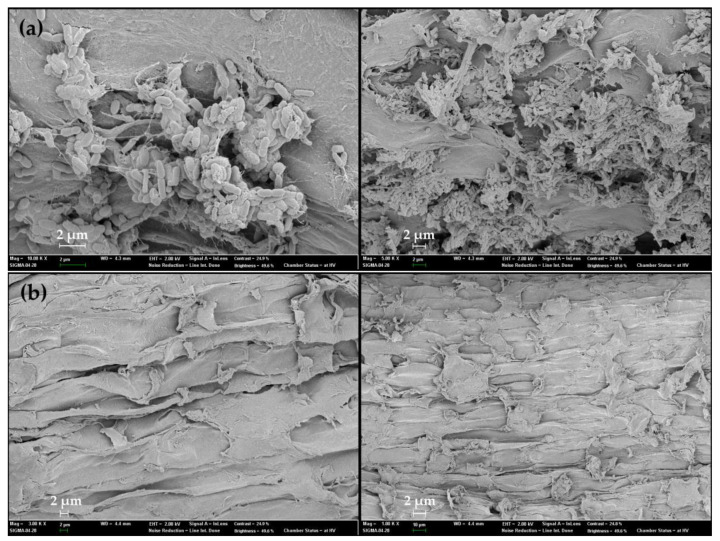
SEM images of rice plant root after 14 days of inoculation. (**a**) root colonized by *Arthrobacter* sp. GN70 and (**b**) control roots without inoculation. Bar 2 μm and 10 μm.

**Figure 7 microorganisms-10-01187-f007:**
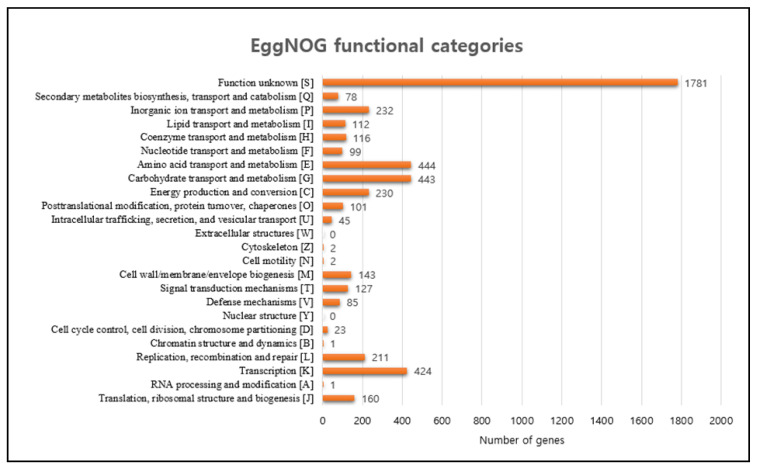
Functional classification of proteins in the genome of strain GN70 by eggNOG annotation.

**Table 1 microorganisms-10-01187-t001:** Sequence analysis of the 16S rRNA gene and plant growth-promoting properties of the 30 bacterial endophytes isolated from the rhizosphere of *Oryza sativa*. L. +, Positive; −, negative.

Sequencing Analysis					Plant Growth Promoting Traits			
No.	Sample Name	Top Taxon Name	16S rRNA Gene Accession No.	Similarity (%)	Siderophore Production	Phosphate Solubilization	Nitrogen Fixation	IAA Production (µg/mL)
1	W1	*Lysobacter niabensis * GH34-4	DQ462461	98.55	+	−	−	5.1
2	S1	*Sphingomonas azotifigens * IFO 15497	NR040835	99.79	+	−	−	8.5
3	S12	*Mycolicibacterium anyangense * QIA-38	KJ855063	100.00	+	−	−	1.9
4	S14	*Streptomyces griseus * KACC 20084	NR042791	100.00	−	−	−	2.8
5	S15	*Stenotrophomonas rhizophila * DSM 14405	NR028930	99.86	+	−	−	8.5
6	Cont1	*Hymenobacter aquatilis * HMF3095	KT273909	100.00	−	−	−	3.8
7	2S11	*Streptomyces novaecaesareae * NBRC 13368	NR041124	99.93	−	−	−	3.9
8	2S17	*Nocardioides terrigena * DS-17	EF363712	99.30	+	−	−	1.8
9	NS3-4	*Microbacterium trichothecenolyticum * DSM 8608	NR044937	99.72	−	−	+	2.6
10	NS3-6	*Caulobacter rhizosphaerae * 7F14	KX792139	99.72	+	−	+	18.2
11	NS3-7	*Sphingomonas mali * IFO 15500	NR026374	100.00	+	−	+	2.6
12	NS5-3	*Alteraurantiacibacter buctensis * M0322	KJ599648	99.65	−	−	+	2.7
13	NS6-1	*Mycolicibacterium llatzerense * MG13	AJ746070	99.28	−	−	+	1.8
14	NS9-1	*Ideonella dechloratans * CCUG 30898	X72724	97.67	+	−	+	3.5
15	NS9-3	*Acidovorax delafieldii* 133	NR028714	99.24	−	−	+	3.6
16	NS11	*Pelomonas saccharophila * DSM 654	NR024710	99.17	−	−	+	8.1
17	NS12-2	*Pelomonas puraquae * CCUG 52769	AM501439	98.96	−	−	+	5.1
18	NS12-5	*Ideonella dechloratans * CCUG 30898	X72724	99.38	+	−	+	4.7
19	P2	*Burkholderia vietnamiensis * LMG 10929	NR041720	100.00	+	+	+	8.9
20	P3	*Burkholderia gladioli * NBRC 13700	AB680484	99.66	+	+	−	6.1
21	P6	*Burkholderia latens * R-5630	AM747628	99.93	+	+	−	7.5
22	P7	*Pelomonas saccharophila * DSM 654	NR024710	98.68	+	−	−	10.6
23	Sa	*Dyella thiooxydans * ATSB10	EF397574	99.18	+	−	−	3.8
24	CT2	*Alteraurantiacibacter buctensis * M0322	KJ599648	99.64	+	−	−	2.4
25	RS2	*Pseudomonas kribbensis * 46-2	KT321658	99.86	+	−	−	10.4
26	RS18	*Agromyces tardus * SJ-23	MH342641	100.00	+	−	−	3.8
27	SO	*Streptomyces fulvissimu * DSM 40593	LM999765	99.86	−	−	−	2.3
28	T2	*Aeromonas veronii * 115/II	NR044845	99.79	−	−	−	4.8
29	T5	*Exiguobacterium undae* L2	AJ344151	99.93	−	−	−	1.7
30	GN70	*Arthrobacter terricola * JH1-1	MG210584	100.00	+	−	+	50.3

**Table 2 microorganisms-10-01187-t002:** Evaluation of antimicrobial activity against pathogenic microorganisms of *Arthrobacter* sp. GN70. All of the experiments were performed on R2A agar plates, except those indicated with a *, which were performed on PDA agar plates. The antimicrobial activity of the following strains was negative: *A. niger* KACC 42850; *B. cinerea* KACC 40573; *C. gloesporioides* KACC 40003; *P. spinosum* KACC 45737; *F. graminearum* KACC 46577; *F. solani* KACC 44891; *F. semitectum* KACC 41036; and *A**. alternata* KACC 44413.

Microorganisms	Zone of Inhibition (Diameter, mm)
*Bacillus subtilis* subsp. *subtilis* KACC 16747	9 ± 1.5
*Staphylococcus aureus* subsp. *aureus* ATCC 6538	16 ± 1.4
*Pantoea agglomerans* KACC 10054	24 ± 0.7
*Xanthomonas campestris* pv. *campestris* KACC 10377	22 ± 2.1
*Pseudomonas aeruginosa* ATCC 6538	16 ± 1.8
*Staphylococcus epidermidis* KACC 13234	18 ± 2.1
** Fusarium proliferatum* KACC 44025	17 ± 1.2

## Data Availability

The data that support the findings of this study are available from the corresponding author upon reasonable request.

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
