# Peer review of "An Isolated *Arthrobacter* sp. Enhances Rice (*Oryza sativa* L.) Plant Growth"

_microorganisms, 2022, doi:10.3390/microorganisms10061187_

Round 1
Reviewer 1 Report
By removing the genomics and problematic figures, most of my criticisms of this manuscript have been addressed.
Author Response
By removing the genomics and problematic figures, most of my criticisms of this manuscript have been addressed.
Ans) Thank you very much for your comments.
Reviewer 2 Report
My comments are listed below.
Lines 88-89: “Root samples were disinfected by washing them three times with distilled water and two times with 70% ethanol” please confirm the disinfection order.
Lines 104-105: “Among the isolates, 60% isolates were able to produce IAA”. It is recommended to replace “60%” with a number.
Line 102-116: This section is repeated with section 3.2. I think this section belonging to results.
Lines172:“Arthrobacter”,please write in italics.
Lines 264-274:Please write the microbe name in italics.
Lines 276: “After 7 and 14 days of incubation……” Please delete the redundant spaces.
Question:
In section 4 discussion, what is meaning of the KACC number with superscript “T”? Why only in this section the KACC number with the superscript?

Author Response
We corrected the whole manuscript according to your instructions in the following ways:
This study revealed that rhizobacterium isolated GN70 from the roots of rice has dual potential to be utilized as a plant growth promoter and antimicrobial agent. As the first report the rice plant growth promoting ability the Arthrobacter species with biofilm producing and a broad spectrum of antimicrobial activities, this is a very exciting results.
My comments are listed below.
Lines 88-89: “Root samples were disinfected by washing them three times with distilled water and two times with 70% ethanol” please confirm the disinfection order.
Ans) We corrected the sentence in lines 98-99.
Lines 104-105: “Among the isolates, 60% isolates were able to produce IAA”. It is recommended to replace “60%” with a number.
Ans) We removed the paragraph, according to your comment below for line 102-116.
Line 102-116: This section is repeated with section 3.2. I think this section belonging to results.
Ans) As you mentioned, we removed the unnecessary part of 2.2, which should be mentioned in 3.2.
Lines172:“Arthrobacter”, please write in italics.
Ans) We corrected in line 182.
Lines 264-274:Please write the microbe name in italics.
Ans) We corrected in lines 292 and 294.
Lines 276: “After 7 and 14 days of incubation……” Please delete the redundant spaces.
Ans) We corrected in line 308.
Question: What is meaning of the KACC number with superscript “T”? Why only in this section the KACC number with the superscript?
Ans) KACC is abbreviation of Korean Agricultural Culture Collection. We already mentioned the abbreviation in lines 175-176. We removed superscript from the discussion in lines 480-501.
This manuscript is a resubmission of an earlier submission. The following is a list of the peer review reports and author responses from that submission.
Round 1
Reviewer 1 Report
In this manuscript, the authors describe the amplicon sequencing and microbial isolation of microbial communities associated with rice roots. Isolates were further tested for their ability to act as growth-promoters for rice biomass, with some genomic characterization of one isolate to identify genes and gene clusters associated with these growth-promoting characteristics.
Overall, this manuscript is very verbose and could be considerably reduced. This includes removing results and discussion-type statements from the introduction and methods, but also includes removing entire sections that don't seem to yield insightful results or strengthen the findings of the manuscript. Particularly, the metagenomic section (not truly "metagenomics" but rather amplicon sequencing) does not have adequate replicates, the findings are not well incorporated with the isolate work, and figures 1 and 2 are too detailed for a reader to see meaningful trends. Additionally, section 3.7 could be removed as there are no quantitative measurements, there are only two plants per treatment, and the growth was cut short at 4 weeks.
Line 24 - Arthrobacter typo
Line 44 - Fertilizers typo
Lines 50-52 - Strange sentences where the "indirect" part ends abruptly, and then continues in the next sentence merged with the "direct" part?
Line 62 - Metagenomics (amplicon sequencing) also has biases and does not get "all" organisms.
Lines 62-76 - The introduction makes it hard to discern what is the original work, and what is the "continuation"
Lines 140-141 - What software and version numbers were used to calculate these alpha-diversity metrics?
Line 145 - Same field as 2.1?
Lines 151, 160 - at least 3 references are missing ("[]").
Line 165 - What does "highest result" mean? Highest of all categories, 30 from each category?
2.4 - Based on what results was Arthrobacter chosen? And there are no methods for how biofilm production was determined.
Line 231 - What was the bacterial concentration of the suspension?
Lines 272-277 - These are measures of richness and not diversity. Phylogenetic diversity (I assume Faith's PD) is not reliable as it is tied to read depth and richness, and is therefore not an independent measure. I would pick one richness metric to report, and remove all others from the table.
Line 282 - These numbers are not the same as Table 1 - which are correct?
Line 299 - What does "purified" mean in this context?
Line 303 - Blast Search results does not explain why 8 isolates were removed from the analysis.
Line 305 - I thought figure 1 was the amplicon taxa, not the isolate taxa?
Line 313 - I find it hard to believe every one of the 504 isolates produce IAA, and I think the methodology may be giving false or misleading results.
Line 364 - Shuffling between PGPB and PGBR - stick with one for consistency.
Line 368 - What does "not much difference" mean? Were any quantitative measurements made or statistics done?
Lines 370-372 - How did an ANOVA give you these percentages?
Line 414 - How was EPS production assayed or confirmed?
Line 463 - Define BGC.
Table 1 - How were "valid reads" determined? Add to methods.
Figure 1 - These are horizontal bars, not vertical. There is also too much detail - a reader cannot discern between, for example, the 9 "blue" taxa. Pick the top 5 and lump all others into an "Others" category.
Figure 2 - Same criticisms as Figure 1, but this time just remove the "Other" entirely as it isn't informative and takes up much of the figure.
Figure 3 - A pie chart with 58 slices is impossible to read/interpret. Remove entirely.
Reviewer 2 Report
This study revealed that rhizobacterium isolated GN70 from the roots of rice has dual potential to be utilized as a plant growth promoter and antimicrobial agent. This strain can contribute to plant healthy growth by biocontrol activity. This is a very exciting results.
But, in my opinion, the experimental design is not rigorous enough. The number of samples in Metagenomic Sequencing are insufficient. Meanwhile, there are lots of mistakes in this manuscript.
My comments are listed below.
lines 39: “Several studies have clearly demonstrated……” please add citations.
lines 148: Fig S1 not in the supplementary.
lines 150-151: “the samples (10 g) were ground using sterile tweezers and a mortar, as described previously []”. The proper citation is missing.
lines 159-160: “The 16S rRNA gene sequencing results were compared in EzBioCloud.net (http://www.ezbiocloud.net) [] and BLAST searches from the NCBI database []”. The proper citation is missing.
lines 171: “centrifugation (16780x g for 5 min)”.please use the correct symbol “×”.
lines 176:“Staphylococcus aureus subsp. aureus”,please correct of the mistakes “aureus” in italic.
lines 218:“Arthrobacter”,please write in italics.
lines 290-291:“Furthermore, sample IL-3 illustrated the highest difference when 290 comparing the 30 most abundant genera among the four samples.”,why not the sample IL-1?
lines 306-307:“Sphingomonas (9%), Streptomyces (4%), Bacillus (4%), Pelomonas (4%), Mycolicibacterium (4%), and Burkholderia (3%) were slightly predominate.”,this result is not consistent with fig 3. I think missed Erythrobacter (4%).
lines 338-339: “GN70 can be utilized as plant growth promoters, suppressors of plant pathogens, and alleviators of water-deficit stress”. Missing necessary experimental results. Please add the “alleviators of water-deficit stress” results.
lines 461-462: “Previous studies showed that phenazine producing bacteria are prevalent in the rhizosphere of rice and wheat [43–45]”, pleas add a period at the end of this sentence.
Fig 8: miss the ruler for the picture, and please add explanations for this four pictures.
lines 430:“[E]”,I think this is not a citation.
lines 440-441: “Arthrobacter species are mostly involve in nitrogen fixation“, please add proper citations.
Some questions:
2.1 “A total of four rice plants were collected from four separate locations” only one plants in each location? I think this experiment need more plants in one location to collect the soil and duplications need design in every location.
2.2 Isolation and Identification. How many fresh roots were collected from one location?
What is the relationship of the samples between 2.1 and 2.2? Does these samples collected from the same location?
2.7. Pot experiment Assay. This experiment use only three seeds in one pot? And no duplication? The plants numbers are too small to represent the results
GN 70 is benefit for rice and can stimulate rice growth as PGPR. There are 9 pathogens were selected to do the antimicrobial and antifungal test. Why not chose more plant or soil pathogens instead of human pathogens?
Based on these considerations this paper is rejected for publication. Considering the value of the found, I suggest the authors to take an adequate period to revise and improve the manuscript and then resubmit it.
